# Haloperidol-Induced Immediate Early Genes in Striatopallidal Neurons Requires the Converging Action of cAMP/PKA/DARPP-32 and mTOR Pathways

**DOI:** 10.3390/ijms231911637

**Published:** 2022-10-01

**Authors:** Oriane Onimus, Emmanuel Valjent, Gilberto Fisone, Giuseppe Gangarossa

**Affiliations:** 1Université Paris Cité, CNRS, Unité de Biologie Fonctionnelle et Adaptative, F-75013 Paris, France; 2Institut de Génomique Fonctionnelle (IGF), Université de Montpellier, CNRS, Inserm, 34094 Montpellier, France; 3Department of Neuroscience, Karolinska Institutet, 17177 Stockholm, Sweden

**Keywords:** haloperidol, striatum, immediate early genes, dopamine, D2R, PKA, mTOR

## Abstract

Antipsychotics share the common pharmacological feature of antagonizing the dopamine 2 receptor (D2R), which is abundant in the striatum and involved in both the therapeutic and side effects of this drug’s class. The pharmacological blockade of striatal D2R, by disinhibiting the D2R-containing medium-sized spiny neurons (MSNs), leads to a plethora of molecular, cellular and behavioral adaptations, which are central in the action of antipsychotics. Here, we focused on the cell type-specific (D2R-MSNs) regulation of some striatal immediate early genes (IEGs), such as cFos, Arc and Zif268. Taking advantage of transgenic mouse models, pharmacological approaches and immunofluorescence analyses, we found that haloperidol-induced IEGs in the striatum required the synergistic activation of A2a (adenosine) and NMDA (glutamate) receptors. At the intracellular signaling level, we found that the PKA/DARPP-32 and mTOR pathways synergistically cooperate to control the induction of IEGs by haloperidol. By confirming and further expanding previous observations, our results provide novel insights into the regulatory mechanisms underlying the molecular/cellular action of antipsychotics in the striatum.

## 1. Introduction

The vast majority of antipsychotics shares the common pharmacological feature of anta-gonizing the dopamine (DA) D2 receptors (D2Rs). These receptors are abundant in the striatum where they are found in half of striatal medium-sized spiny neurons (MSNs) and in cholinergic interneurons but also on terminals of projecting DA-neurons [1,2].

D2R-MSNs represent the main cell type through which antipsychotics (i.e., haloperidol, raclopride, eticlopride, sulpiride) can lead to extrapyramidal and motor side effects such as hypolocomotion, catalepsy, parkinsonism and tardive dyskinesia [3,4]. A large body of evidence indicates that D2R antagonists, by blocking/preventing the DA→D2R→Gi (inhibitory G protein-coupled receptor)-mediated tonic inhibition of D2R-MSNs, promote the activation/disinhibition of intracellular signaling events through the recruitment of multiple pathways, including the cAMP-dependent protein kinase A (PKA)/dopamine- and cAMP-regulated phosphoprotein (DARPP-32) and the mTOR cascades [5,6,7,8,9], which participate in the regulation of both transcriptional and translational modifications. However, although blockade of the endogenous action of DA onto D2R is necessary to promote the activation of signaling cascades in D2R-MSNs, several reports have also shown that D2R downstream intracellular signaling events (i.e., cAMP/PKA pathway) also depend on the dynamic orchestration of several extracellular and intracellular players whose activity may scale the modulation of D2R-containing neurons [7,8,9,10,11,12,13], thus reflecting the nature of MSNs as coincidence detectors of different neurochemical stimuli. Within the striatum, this antipsychotics-associated signaling cascade rapidly culminates in the long-term expression of immediate early genes (IEGs), such as cFos, Arc and Zif268 [14].

IEGs represent a heterogenous class of highly responsive genes that can be rapidly and transiently induced by a plethora of physiological, pharmacological and environmental stimuli. The encoded proteins of IEGs cFos, Arc and Zif268 are well known to contribute to forms of synaptic plasticity [15]. Indeed, within the striatum, the study of DA transmission, DA-mediated signaling events and IEGs has been instrumental in shedding lights on the temporal and spatial organization of striatal territories as well as on the functions of striatal cell types, circuits and neuromodulatory systems [16,17,18,19,20,21,22,23,24,25]. However, whether and how multiple (extracellular and intracellular) signaling pathways converge on the regulation of haloperidol-induced IEGs in D2R-MSNs remain to be fully understood.

Here, by using transgenic animals, pharmacological approaches and immunofluorescence analyses, we investigated whether and how (i) non-DA neurotransmitters, notably glutamate (via NMDAR) and adenosine (via A2aR), and (ii) intracellular PKA/DARPP-32 and mTOR signaling pathways contribute to the regulation of haloperidol-induced IEGs in the striatum. We observed that NMDAR and A2aR as well as PKA/DARPP-32 and mTOR pathways are distinctly required and synergistically necessary to gate haloperidol-induced activation of D2R-MSNs. Our study, by further expanding the landscape of cellular and molecular modifications elicited by antipsychotics, provide new evidence for the regulatory activity of D2R-MSNs.

## 2. Results

### 2.1. Haloperidol Induces Contrasting Cell Type-Specific Regulation of IEGs in the Dorsal Striatum

It is well established that haloperidol induces the expression of the IEGs cFos, Arc and Zif268 in the striatum [26,27,28,29,30,31]. Here, we took advantage of *Drd2*-eGFP mice [32] to visualize striatopallidal D2R^+^-neurons and revisit the cell type-specific expression patterns of cFos, Arc and Zif268 induced by haloperidol. As expected, an increase in cFos-, Arc- and Zif268-immunoreactive neurons was observed in the dorsal striatum 60 min after a single administration of haloperidol (0.5 mg/kg, ip) (Figure 1A).

Cell type-specific analysis revealed that cFos and Arc were exclusively triggered in striatopallidal D2R^+^-neurons (Figure 1A,B). In contrast, Zif268 was increased in both D2R^-^-neurons (putative striatonigral neurons) and striatopallidal D2R^+^-neurons (Figure 1A,B), although Zif268^+^/D2R^+^-neurons outnumbered Zif268^+^/D2R^-^-neurons (Figure 1A,B). This first set of results indicates that cFos and Arc, but not Zif268, can be used as bonafide markers to study long-term cellular and molecular events induced by haloperidol in striatopallidal neurons.

Next, we wondered whether the induction of cFos and Arc spatially occurred in the same or different subpopulations of D2R^+^-neurons. Thus, we performed a double immunofluorescence analysis for cFos and Arc. We observed a complete colocalization (341 out of 341 neurons, 100%) of cFos with Arc (Figure 2A,B). However, we also observed that among all Arc^+^-neurons, 23.5% (105 out of 446 neurons) did not express cFos (Figure 2A,B), therefore indicating that haloperidol-induced cellular modifications may occur at different spatial scales.

### 2.2. Haloperidol-Induced cFos and Arc Require the Activation of A2a and NMDA Receptors

Next, we investigated the contribution of non-DA receptors in the regulation of haloperidol-induced cFos and Arc (Figure 1). We focused on A2a (highly enriched in striatopallidal neurons [33]) and NMDA receptors as they strongly contribute to the regulation of the striatopallidal neurons’ activity and plasticity [34,35].

We observed that pretreatment with the specific A2aR antagonist KW-6002 (3 mg/kg [8]) significantly reduced haloperidol-induced cFos expression in the striatum (Figure 3A–C). In addition, when mice were pretreated with the NMDAR antagonist MK-801 (0.1 mg/kg), we noticed a dramatic reduction in haloperidol-induced cFos activation (Figure 3A–C). However, both A2aR and NMDAR blockades failed in fully preventing cFos expression in haloperidol-treated mice. Interestingly, the co-administration of KW-6002 and MK-801 before haloperidol injection resulted in a complete loss of cFos activation (Figure 3A–C). Similar regulatory mechanisms were observed when Arc was used as a molecular proxy of the haloperidol-induced activation of striatopallidal neurons. In fact, while KW-6002 and MK-801 distinctly reduced haloperidol-induced Arc expression (Figure 3D,E), the co-administration of both A2aR and NMDAR antagonists completely prevented the cellular/molecular response of striatopallidal neurons to haloperidol (Figure 3D,E).

These results indicate that ambient adenosine and glutamate are necessary and act synergistically to gate the responsiveness of striatopallidal neurons to the antipsychotic haloperidol.

### 2.3. Haloperidol-Induced cFos and Arc Require the Involvement of PKA/DARPP-32 and mTOR Pathways

Haloperidol promotes the PKA-dependent phosphorylation of DARPP-32 on Thr34, which converts DARPP-32 into an inhibitor of protein phosphatase 1 (PP-1), thereby inhibiting the dephosphorylation of numerous cAMP/PKA-dependent molecular targets [36]. Indeed, at the intracellular level, A2aR and D2R oppositely regulate the cAMP/PKA/DARPP-32 pathway [7], and seminal reports have shown that the cellular modifications elicited by haloperidol depend on the recruitment of key intracellular signaling cascades such as the PKA/DARPP-32 and mTOR pathways [7,8,9,37]. Here, we decided to investigate whether these two pathways were, distinctly and/or synergistically, involved in the regulation of haloperidol-induced IEGs.

To dampen the PKA/DARPP-32 striatal path in vivo, we used DARPP-32 mutant mice in which Thr34 was replaced by Ala (T34A) [38]. Compared to WT mice, T34A mutant mice showed reduced cFos and Arc activation following haloperidol administration (Figure 4A–E). To inhibit the mTOR signaling pathway, mice were pretreated with rapamycin (5 mg/kg, [9]). As shown in Figure 4, rapamycin reduced haloperidol-induced cFos and Arc (Figure 4A–E). Interestingly, when both PKA/DARPP-32 and mTOR pathways were downregulated (rapamycin administration to T34A mutant mice), we observed a more pronounced reduction in cFos and Arc following haloperidol administration.

These results indicate that intracellular cAMP/PKA and mTOR pathways are both synergistically mobilized and required for the full cellular/molecular action of haloperidol onto striatopallidal neurons.

## 3. Discussion

In this study we show that the typical antipsychotic haloperidol, which alters the activity and plasticity of D2R-MSNs [39,40], elicits cell type-specific activation of IEGs in the mouse striatum (D1R-MSNs vs. D2R-MSNs) and that this activation requires the contribution of different signaling actors. Here, we focused on cFos, Arc and Zif268 which, by representing the result of long-term cellular/molecular adaptations, are dynamically triggered by a plethora of stimuli and are also involved in the regulation of several neuronal functions [41,42,43,44]. We observed that the haloperidol-induced expression of the IEGs cFos and Arc was exclusively restricted to striatopallidal D2R^+^-neurons, whereas Zif268 expression was triggered in both D2R^-^-neurons (putative striatonigral D1R-MSNs) and striatopallidal neurons (D2R-MSNs). These findings are in line with and further extend previous observations on haloperidol-regulated phospho-proteins (histone H3, ribosomal protein S6), highlighting the cell type-specific reactivity of MSNs to DA-related agents [7,8,18,19]. However, the cell type-unspecific expression of Zif268 in the striatal MSNs suggests that haloperidol, most likely through an indirect mechanism involving striatal cholinergic and/or GABAergic interneurons [45,46], may also regulate the activity of D1R-MSNs even though no major changes in the electrophysiological profile of this cell type have been observed [40]. In this study, we focused on the rostral segment of the dorsal striatum. However, the striatum extends throughout a rostro-caudal axis with major differences in the topographic and functional organization of striatal domains and cell types [17,22,24,47,48]. Indeed, it will be of high interest to explore whether and how antipsychotics impact on the regulatory functions of caudal striatal domains.

In addition to the selective induction of cFos and Arc in striatopallidal neurons following haloperidol administration, we noticed that cFos^+^/D2R^+^-neurons seemed to represent a subpopulation of Arc^+^/D2R^+^-neurons. This is of interest in light of recent reports describing that the two major striatal cell populations (D1R- and D2R-MSNs) may actually consist of different subpopulations with distinct spatiocellular and spatiomolecular features [49,50,51,52]. Indeed, the use of unbiased high-throughput technologies, either at the single cell or population level, will be instrumental to reappraise the functional heterogeneity of striatal neurons.

The therapeutic action as well as the side effects of antipsychotics strongly depend on the pharmacological blockade of D2R. However, other neurotransmitters participate in regulating the cellular activity of D2R-containing neurons and consequently scale the behavioral effects of haloperidol (i.e., catalepsy). Adenosine and glutamate represent two of the major neuromodulators/neurotransmitters able to regulate the activity of MSNs, and seminal reports have shown that their respective receptors, A2aR and NMDAR, distinctly influence haloperidol-associated effects. In fact, the pharmacological administration of A2aR antagonists and/or genetic deletion of A2aR counteract haloperidol-associated behaviors [53,54,55]. This is in line with our observation showing that the blockade of A2aR was sufficient to partially blunt haloperidol-induced cFos and Arc expression in the striatum. This action is mainly due to the opposite regulation exerted on D2R-MSNs by A2aR and D2R, which are positively (via G_s_) and negatively (via G_i_) coupled to the adenylyl cyclase, respectively. While the A2aR and D2R downstream signaling events may buffer the intracellular production of cAMP, several reports have also indicated that A2aR and D2R can heterodimerize (protein–protein interaction) at the membrane levels, therefore influencing the activity of each receptor [56,57,58,59,60,61].

In addition to ambient striatal adenosine, the excitatory neurotransmitter glutamate, mostly released from cortico-striatal and thalamo-striatal projections, is another key player in the regulation of MSNs. Seminal studies have shown that the pharmacological blockade of NMDAR was sufficient to blunt haloperidol-induced catalepsy [62,63,64], and our study provides further evidence that NMDARs are required for haloperidol-induced cFos [65,66] and Arc in the striatum. Indeed, NMDARs can gate the action of haloperidol (D2R) through multiple mechanisms. First, the activation of NMDAR can boost the activation of the cAMP/PKA cascade via an indirect A2aR-dependent mechanism [67]. Second, haloperidol-induced cAMP/PKA activity can lead to the PKA-dependent phosphorylation of NMDAR subunits, thus facilitating glutamatergic transmission [68]. Third, recent reports have shown that dopamine receptors can also heterodimerize with NMDARs [69,70,71]. Fourth, the blockade of NMDAR may also counteract the D2R-dependent modulation of cortico-striatal glutamate transmission [72,73]. Therefore, NMDAR antagonists may blunt haloperidol-elicited striatal IEGs through all these mechanisms.

However, our study also shows that while the blockade of A2aR or NMDAR blunted haloperidol-induced IEGs, the co-administration of A2aR and NMDAR antagonists (KW-6002 and MK-801, respectively) resulted in the full abolishment of cFos and Arc. These results indicate that to modulate striatal networks, haloperidol requires the dynamic and synergistic action of dopamine (D2R), adenosine (A2aR) and glutamate (NMDAR).

At the intracellular molecular level, haloperidol mobilizes the PKA/DARPP-32 cascade as well as the mTOR signaling pathway [7,8,9,37,74,75]. Indeed, the downregulation of the PKA/DARPP-32 [76] and mTOR [75] pathways dampened haloperidol-induced catalepsy. In line with these reports, we observed a reduction in cFos and Arc when haloperidol was administered to DARPP-32 T34A mutant mice (inhibition of PKA/DARPP-32 cascade) and also when it was preceded by the injection of rapamycin (inhibition of mTOR). However, we also noticed that when both signaling paths were downregulated (rapamycin administration to T34A mice), haloperidol completely failed to induce cFos and Arc expression in the striatum, therefore indicating that both signaling cascades are necessary and required for the full action of haloperidol. It is interesting to note that beside the antagonistic functions of A2aR and D2R on the cAMP/PKA/DARPP-32 pathway, NMDAR have also been associated to the downstream activation of the mTOR pathway [77,78,79], thus indicating that the converging contribution of these two intracellular signaling cascades may actually reflect the dynamic involvement of extracellular adenosine and glutamate in mediating the D2R-dependent action of haloperidol in striatopallidal neurons. Indeed, our results describing the dynamic regulation of striatal IEGs represent also a call for the use of modern high-throughput quantitative technologies to deeply investigate the cell type-specific adaptations elicited by antipsychotics.

In conclusion, our study, by confirming and expanding previous observations and also by providing a new synergic perspective of striatal regulatory mechanisms, highlights the heterogenous complexity of in vivo processes which may be involved in the therapeutic and adverse effects of antipsychotics.

## 4. Material and Methods

### 4.1. Animals

For all experiments, 8–12-week-old mice were used. Male C57BL/6J mice (25–30 g) were purchased from Taconic (Tornbjerg, Denmark). Bacterial artificial chromosome transgenic mice expressing eGFP under the control of the promoter for the D2R (*Drd2*-eGFP) were generated by the GENSAT program (Gene Expression Nervous System Atlas) at the Rockefeller University [80] and backcrossed on a C57BL/6J background. Knock-in mice expressing a mutated form of DARPP-32, in which the threonine (Thr34) is replaced by an alanine (DARPP-32 T34A mutant mice), were generated as previously described [38] and backcrossed on a C57BL/6J background. Animals were maintained in a 12 h light–dark cycle at a stable temperature of 22 °C, with food and water ad libitum. To reduce the stressogenic component of in vivo manipulations, before any pharmacological experimentation, mice were handled and injected with saline for three consecutive days. The experiments were conducted in accordance with the guidelines of the Research Ethics Committee of Karolinska Institutet, Swedish Animal Welfare Agency and the 2010/63/EU directive for the care and use of experimental animals.

### 4.2. Drugs

Haloperidol (0.5 mg/kg, Sigma-Aldrich, St. Louis, MO, USA) was dissolved in saline containing 0.05% (*v*/*v*) acetic acid with the pH adjusted to 6.0 with 1 M NaOH, and it was injected 60 min before perfusion. The A2aR antagonist KW-6002 (also known as istradefylline, 3 mg/kg, [8]), a gift from Dr. Edilio Borroni (Hoffmann-LaRoche, Basel, Switzerland), was suspended by sonication in a solution of 5% (*v*/*v*) Tween 80 in saline and administered 5 min before haloperidol as previously reported [8]. The NMDAR antagonist MK-801 (also known as dizocilpine, 0.1 mg/kg, Tocris, Bristol, UK) was dissolved in saline and administered 30 min before haloperidol, which is in line with previous reports [17,81,82]. Rapamycin (5 mg/kg, LC Laboratories, Woburn, MA, USA) was dissolved in a solution of 5% (*v*/*v*) dimethylsulfoxide (DMSO), 5% Tween-20 and 15% (*v*/*v*) PEG-400, and administered (once per day) starting 3 days before the experiment. On the day of the experiment, rapamycin was injected 45 min before haloperidol as previously reported [9]. All drugs were administered intraperitoneally (i.p.) in a volume of 10 mL/kg, except for rapamycin (mTOR inhibitor), which was administered in a volume of 5 mL/kg. When mice were not treated with drugs, they received an equivalent volume of the corresponding vehicle.

### 4.3. Tissue Preparation and Immunofluorescence

Tissues were prepared as previously described [17]. In brief, mice were rapidly anaesthetized by i.p. injection of pentobarbital (500 mg/kg, Sanofi-Aventis, Paris, France) prior to intracardiac perfusion of 4% (*w*/*v*) paraformaldehyde (PFA) in 0.1 M Na_2_HPO_4_/NaH_2_PO_4_ buffer (pH 7.5), which was delivered with a peristaltic pump at 20 mL/min over 5 min. Brains were post-fixed overnight in the same solution and stored at 4 °C.

Thirty μm thick sections were cut with a vibratome (Leica, France) and stored at −20°C in a solution containing 30% (*v*/*v*) ethylene glycol, 30% (*v*/*v*) glycerol, and 0.1 M sodium phosphate buffer until they were processed for immunofluorescence.

Sections were processed as follows: free-floating sections were rinsed three times for 10 min in Tris-buffered saline (TBS, 50 mM Tris–HCL, 150 mM NaCl, pH 7.5). After 15 min of incubation in 0.2% (*v*/*v*) Triton X-100 in TBS, sections were rinsed in TBS again and blocked for 1 h in a solution of 3% BSA in TBS. Sections were then incubated for 72 h at 4 °C in 1% BSA, 0.15% Triton X-100 with the primary antibodies. Antibodies for cFos (1∶400, #sc-52), Arc (1∶400, #sc-17839), and Zif268 (1∶400, #sc-189) were purchased from Santa Cruz Biotechnology [83]. GFP was amplified by using a chicken anti-GFP (1:1000, Invitrogen/ThermoFisher Scientific, Waltham, MA, USA). Sections were rinsed three times for 10 min in TBS and incubated for 45 min with goat Cy3-, Cy5-coupled (1:400, Jackson Immunoresearch, Cambridge House, St. Thomas’ Place, UK), and/or goat A488 (1:400, Invitrogen/ThermoFisher Scientific, Waltham, MA, USA) secondary antibodies. Sections were rinsed for 10 min twice in TBS and twice in Tris-buffer (1 M, pH 7.5) before mounting in a 1,4-diazabicyclo-[2.2.2]-octane (DABCO, Sigma-Aldrich) solution.

### 4.4. Confocal Microscopy and Image Analysis

Single and/or double-immunolabeled images from each region of interest (dorsal striatum) were obtained using sequential laser scanning confocal microscopy (Zeiss LSM510 META, Oberkochen, Germany). Images were acquired at the level of the dorsal striatum (rostral level). Photomicrographs were obtained with the following band-pass and long-pass filter setting: GFP (band pass filter: 505–530), Cy3 (band-pass filter: 560–615) and Cy5 (long-pass filter 650).

Quantifications were performed in 325.75 µm × 325.75 µm confocal images. Immunofluorescent striatal cells (expressing IEGs and/or eGFP) were counted (absolute number) blindly of the treatment/group by using the cell counter plugin of the ImageJ software taking as standard reference a fixed threshold of fluorescence. Of note, the distinction between striatal cell types (Figure 1) was performed using *Drd2*-eGFP mice (positive vs. negative neurons) [32,80].

### 4.5. Statistical Analysis

All statistical comparisons were performed with two-sided tests in Prism 6 (GraphPad Software, La Jolla, CA, USA). The distribution of data was determined with a Shapiro–Wilk normality test. No sample size calculations were performed. To determine outliers in every experimental group, we performed the Grubbs’ test in Prism. No outliers were identified. All the data were analyzed using either Student’s *t*-test with equal variances or one-way ANOVA. In all cases, the significance threshold was automatically set at *p* < 0.05. The one-way ANOVA analysis, where treatment was the independent variable, was followed by a Bonferroni post hoc test for specific comparisons only when the overall ANOVA revealed a significant difference (at least *p* < 0.05) among groups. All the values and statistical analyses are reported in the figure legends.

## Figures and Tables

**Figure 1 ijms-23-11637-f001:**
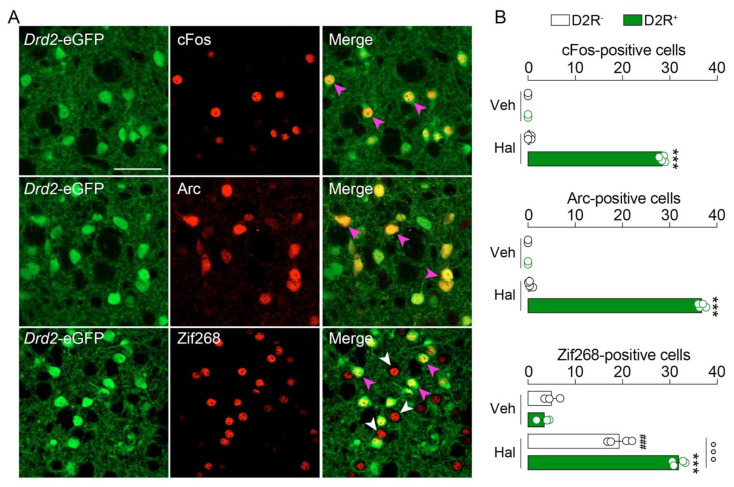
Haloperidol induces cFos, Arc and Zif268 in a cell type-specific manner in the mouse striatum. (**A**) Double immunofluorescence detection of cFos, Arc and Zif268 in the striatum of *Drd2*-eGFP mice 60 min after vehicle (Veh) or haloperidol (Hal) administration. Arrows in magenta indicate the expression of immediate early genes (IEGs) in D2R^+^-neurons, whereas arrows in white indicate the expression of IEGs in D2R^-^-neurons. Scale bar: 50 μm. Note: given the low number of IEGs-positive neurons in Veh-treated mice, pictures are not shown. (**B**) Quantification of cFos, Arc and Zif268 in both D2R^+^- and D2R^−^-neurons. Statistics: *** *p* < 0.001 (Hal vs. Veh in D2R^+^-neurons), ^###^ *p* < 0.001 (Hal vs. Veh in D2R^−^-neurons), °°° *p* < 0.001 (Hal in D2R^+^-neurons vs. Hal in D2R^−^-neurons for Zif268). Veh (*n* = 3) and Hal (*n* = 4). One-way ANOVA: F_(3, 10)_ = 5301, *p* < 0.0001 (cFos); F_(3, 10)_ = 5618, *p* < 0.0001 (Arc); F_(3, 10)_ = 180.7, *p* < 0.0001 (Zif268).

**Figure 2 ijms-23-11637-f002:**
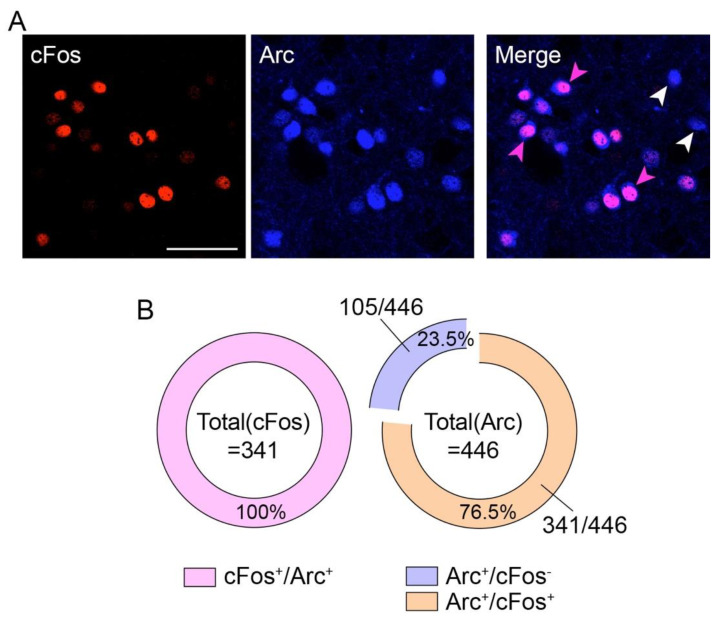
Spatiocellular organization of haloperidol-induced cFos and Arc. (**A**) Double immunofluorescence detection of cFos (red) and Arc (blue) following haloperidol administration. Scale bar: 50 μm. Arrows in magenta indicate the co-expression of cFos and Arc in the same neurons, whereas arrows in white indicate the expression of Arc (no cFos co-expression). (**B**) Quantification of the degree of colocalization between cFos and Arc. Note that 23.5% of Arc^+^-neurons did not express cFos.

**Figure 3 ijms-23-11637-f003:**
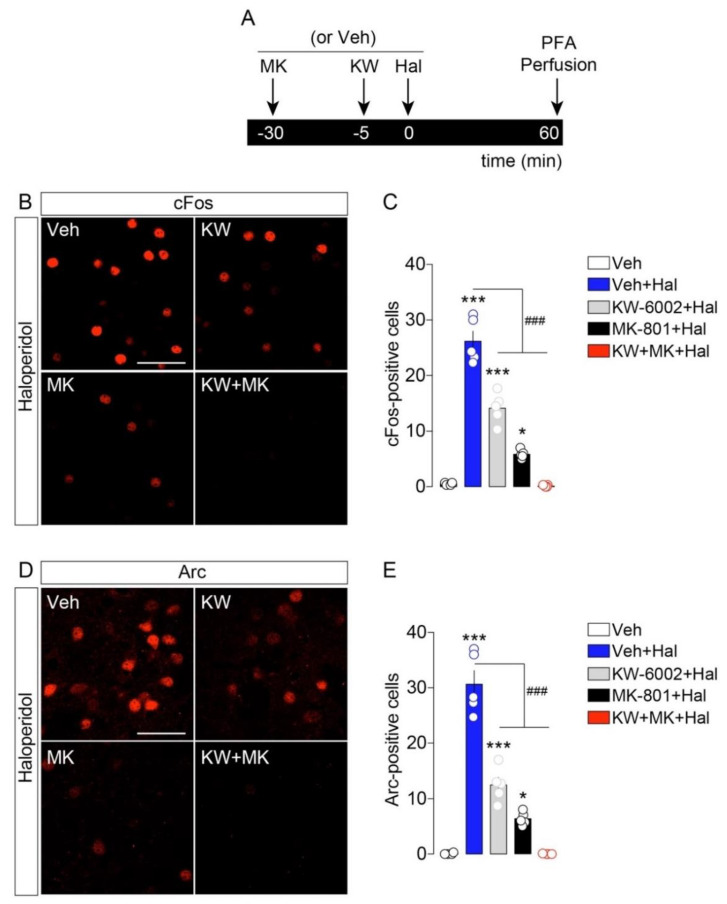
Haloperidol-induced cFos and Arc require the activation of A2a and NMDA receptors. (**A**) Temporal schedule of pharmacological treatments. (**B**) Immunofluorescence detection of cFos in the striatum of animals treated with vehicle + haloperidol (*n* = 5), KW-6002 (KW) + haloperidol (*n* = 5), MK-801 (MK) + haloperidol (*n* = 5) and KW-6002 (KW) + MK-801(MK) + haloperidol (*n* = 5). Note: given the low number of IEGs-positive neurons in Veh-treated mice, pictures are not shown. Scale bar: 50 μm. (**C**) Quantification of cFos^+^-neurons in all experimental groups. Statistics: * *p* < 0.05 and *** *p* < 0.001 (drugs vs. vehicle), ^###^ *p* < 0.001 (KW-6002 + haloperidol and MK-801 + haloperidol vs. haloperidol). One-way ANOVA: F_(4, 19)_ = 113.9, *p* < 0.0001. (**D**) Immunofluorescence detection of Arc in the striatum following pharmacological administrations. Scale bar: 50 μm. (**E**) Quantification of Arc^+^-neurons. Statistics: * *p* < 0.05 and *** *p* < 0.001 (drugs vs. vehicle), ^###^ *p* < 0.001 (KW-6002 + haloperidol and MK-801 + haloperidol vs. haloperidol). One-way ANOVA: F_(4, 19)_ = 90.46, *p* < 0.0001.

**Figure 4 ijms-23-11637-f004:**
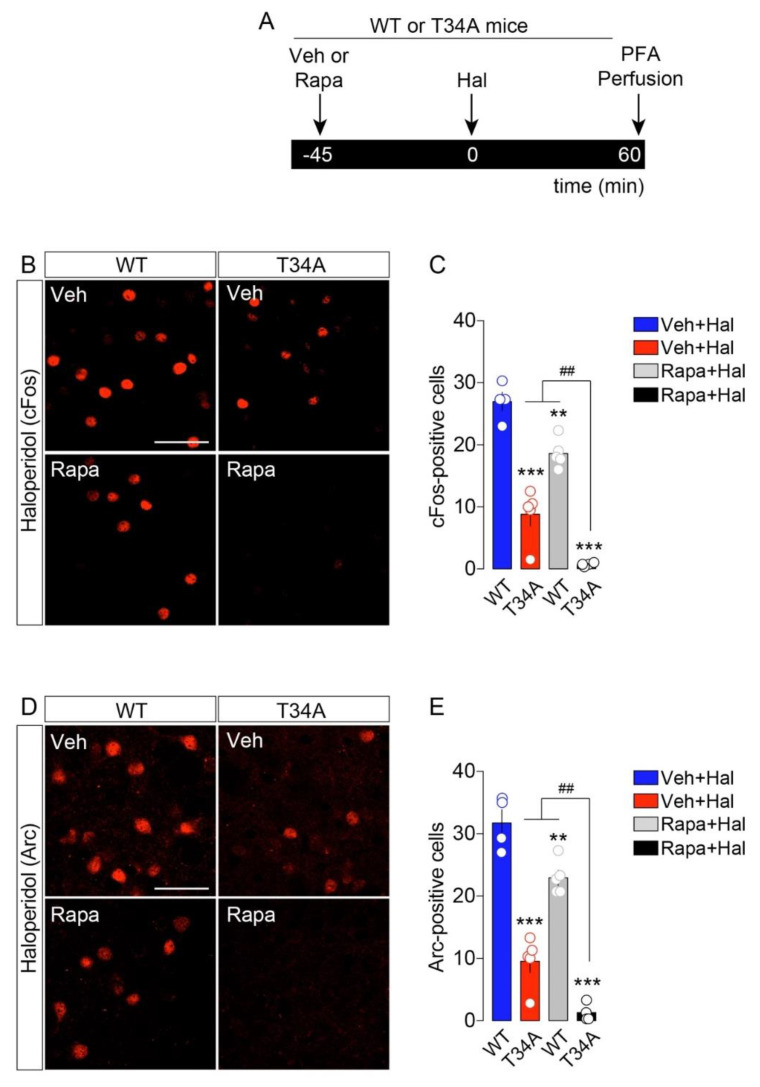
Haloperidol-induced cFos and Arc require the intracellular activation of PKA/DARPP-32 and mTOR pathways. (**A**) Temporal schedule of pharmacological treatments. (**B,D**) Immunofluorescence detection of cFos (**B**) and Arc (**D**) in the striatum of WT and T34A animals treated with vehicle + haloperidol, or rapamycin + haloperidol. Scale bars: 50 μm. (**C**) Quantification of cFos^+^-neurons in WT and T34A mutant mice treated with vehicle + haloperidol and rapamycin + haloperidol (*n* = 4–5/group). Statistics: ** *p* < 0.01 and *** *p* < 0.001 (T34A + Hal, WT + Rapa + Hal, T34A + Rapa + Hal vs. WT + Hal), ^##^ *p* < 0.01 (T34A + Rapa + Hal vs. T34A + Hal and WT + Rapa + Hal). One-way ANOVA: F_(3, 14)_ = 62.67, *p* < 0.0001. (**E**) Quantification of Arc^+^-neurons in WT and T34A mutant mice treated with vehicle + haloperidol and rapamycin + haloperidol (*n* = 4–5/group). Statistics: ** *p* < 0.01 and *** *p* < 0.001 (T34A + Hal, WT + Rapa + Hal, T34A + Rapa + Hal vs. WT + Hal), ^##^ *p* < 0.01 (T34A + Rapa + Hal vs. T34A + Hal and WT + Rapa + Hal). One-way ANOVA: F_(3, 14)_ = 70.07, *p* < 0.0001.

## Data Availability

The data presented in this study are available in the Figures and Results sections.

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
