# Peer review of "Haloperidol-Induced Immediate Early Genes in Striatopallidal Neurons Requires the Converging Action of cAMP/PKA/DARPP-32 and mTOR Pathways"

_ijms, 2022, doi:10.3390/ijms231911637_

Round 1

Reviewer 1 Report

The study examined the expression of several immediate early genes in striatopallidal neurons when exposed to an antipsychotic drug (haloperidol). The obtained data confirm and expand the previously published observations of other authors.

Comments that require correction of the text of the manuscript before publication are listed below.

Line 3: converging activation – activation is not shown, it is better to write expression

 Line 18: “synergistic activation of” - activation is not shown, it is better to write expression

Line 34: Gi-mediated – need to be deciphered

 Lines 68-69: Male C57BL/6J mice (25–30 g) were purchased from Taconic (Tornbjerg, Denmark), -  it is not clear why these mice are indicated, they were not used in the experiment.

 Line 77: “mice were handled and injected with saline for three consecutive days” – it is necessary to explain to readers why they did it.

 Line 84: KW6002 – it is necessary to describe what it is and why it was used.

Line 86: MK-801 - it is necessary to describe what it is and why it was used..

 Lines 82-94: a drawing is required to graphically explain the scheme of experiments indicating the doses, time intervals for the administration of drugs and the animals used. It is not clear why different blockers were administered in different schedules (5 or 30 minutes before haloperidol). It is necessary to substantiate the chosen temporary regimens for the administration of the used blockers and the chosen regimen for the administration of rapamycin.

Lines 142-143: an increased number – it is not clear in comparison with which control one can see an increase in the number of immunoreactive neurons.

Line 146: it is not clear why the authors believe that Zif268 was increased in both striatonigral (D2R--neurons) and striatopallidal D2R+-neurons. Figure 1 shows that it was expressed in both types of neurons. To say that the expression of Zif268 was increased, it is necessary to show the pattern of the negative control (ie, without induction with haloperidol).

Lines 176-177:” A2aR antagonist KW-6002 [also known as istradefylline, 3 mg/kg, [8]]” – This is information for the Methods section.

 Line 179: “NMDAR antagonist MK-801 (also known as dizocilpine, 0.1 mg/kg)” ]]” – This is information for the Methods section.

 Line 193: “the activation of A2a and NMDA receptors” – it is better to replace the word activation with expression.

 Line 194: : ”(A) Immunofluorescence detection of cFos in the striatum of animals treated with vehicle (n=4),” This is not shown in Figure 3a. Figure 3C also shows only 4 figures, and the interpretation (Figure 3d) is given for five options. You need to either add pictures or give an explanation, for example (not shown).

 Line 230: Halo, - write uniformly, please

 Line 219: “we observed a full abolishment in the induction of cFos and Arc following haloperidol administration” – this is not entirely correct, since according to Figure 4D Rapa + Hal is not equal to zero

 Line 243: “haloperidol-induced activation of the IEGs cFos and Arc” – activation was not shown in this study. No negative control pattern was given. Either pictures are needed, or links to articles in which this is shown.

Author Response

The study examined the expression of several immediate early genes in striatopallidal neurons when exposed to an antipsychotic drug (haloperidol). The obtained data confirm and expand the previously published observations of other authors.

We would like to thank the reviewer for his/her inputs.

Comments that require correction of the text of the manuscript before publication are listed below.

Line 3: converging activation – activation is not shown, it is better to write expression

We have changed activation into action.

Line 18: “synergistic activation of” - activation is not shown, it is better to write expression

This is an activation since with A2aR and NMADAR antagonists we block the endogenous action of adenosine (A2aR) and glutamate (NMDAR)

Line 34: Gi-mediated – need to be deciphered

Done (line 34)

Lines 68-69: Male C57BL/6J mice (25–30 g) were purchased from Taconic (Tornbjerg, Denmark), -  it is not clear why these mice are indicated, they were not used in the experiment.

These mice were used for the pharmacological studies of Fig. 3.

Line 77: “mice were handled and injected with saline for three consecutive days” – it is necessary to explain to readers why they did it.

This was done to reduce stress. We have mentioned this point in line 77-78.

Line 84: KW6002 – it is necessary to describe what it is and why it was used.

Corrected (line 86)

Line 86: MK-801 - it is necessary to describe what it is and why it was used.

Corrected (line 89)

Lines 82-94: a drawing is required to graphically explain the scheme of experiments indicating the doses, time intervals for the administration of drugs and the animals used. It is not clear why different blockers were administered in different schedules (5 or 30 minutes before haloperidol). It is necessary to substantiate the chosen temporary regimens for the administration of the used blockers and the chosen regimen for the administration of rapamycin.

Thank you for this point.

Time schedules and doses of drugs were adopted according to previous reports mentioned in the manuscript which have clearly shown their efficacity in vivo. As requested by the reviewer, we have included a graphical representation of the pharmacological experiments.

Lines 142-143: an increased number – it is not clear in comparison with which control one can see an increase in the number of immunoreactive neurons.

This has been corrected. We now also show data from vehicle-treated mice, clearly indicating the increase with proper comparisons (Fig. 1).

Line 146: it is not clear why the authors believe that Zif268 was increased in both striatonigral (D2R--neurons) and striatopallidal D2R+-neurons. Figure 1 shows that it was expressed in both types of neurons. To say that the expression of Zif268 was increased, it is necessary to show the pattern of the negative control (ie, without induction with haloperidol).

Please see new Fig. 1.

Lines 176-177:” A2aR antagonist KW-6002 [also known as istradefylline, 3 mg/kg, [8]]” – This is information for the Methods section.

Corrected

 Line 179: “NMDAR antagonist MK-801 (also known as dizocilpine, 0.1 mg/kg)” ]]” – This is information for the Methods section.

Corrected

 Line 193: “the activation of A2a and NMDA receptors” – it is better to replace the word activation with expression.

Adenosine and glutamate activate these 2 receptors, expression is related to IEGs.

 Line 194: : ”(A) Immunofluorescence detection of cFos in the striatum of animals treated with vehicle (n=4),” This is not shown in Figure 3a. Figure 3C also shows only 4 figures, and the interpretation (Figure 3d) is given for five options. You need to either add pictures or give an explanation, for example (not shown).

Thank for this point which has been corrected. We did not show the picture since vehicle mice do not show induced Arc and cFos.

 Line 230: Halo, - write uniformly, please

Corrected

 Line 219: “we observed a full abolishment in the induction of cFos and Arc following haloperidol administration” – this is not entirely correct, since according to Figure 4D Rapa + Hal is not equal to zero

Rephrased and corrected

Line 243: “haloperidol-induced activation of the IEGs cFos and Arc” – activation was not shown in this study. No negative control pattern was given. Either pictures are needed, or links to articles in which this is shown.

Corrected.

Reviewer 2 Report

The authors study the Haloperidol-induced immediate early genes in striatopallidal neurons with specific A2A and NMDA receptors relevant and via mTOR and PKA pathways. These investigations should be interesting.

However, some major defects exist:

1, the authors only provide some IHC images as well as quantitative analysis based on IHC data. These observations are largely not enough to make solid conclusions. Western blot study and quantitative real time RT-PCR of IEG gene expression changes should be more convincing. Therefore the authors are request to perform western blot analysis and RT-PCR to ensure all IEG gene expression after Haloperidol treatment.

2, The authors only observe some phenotype changes of Haloperidol treatment, such as A2A and NMDA receptors relevant and PKA, mTOR pathway relevant. However, what the logical links among these observations. So the study is very superficial with little impactful and novel knowledge. The potential impact of current study is very limited.

 3, there are many defects on experimental design:

(1), in Figure 1, the authors should stain the tissue with hoechst so as to identify how many cells exposed in the image (D2R+ and D2R-). How the authors count the total D2R- cells? There are quite lots of black color holes in the images, which should be the cells without staining by red and green color. If only count the positive stained cells, that will be misleading. This is because the total numbers of D2R+ and D2R- cells are not the same. The author should study the percentage of red color cells of D2R+ and D2R- cells.

(2), in Figure 4, the authors should study the phosphorylation status of downstream proteins of mTOR and PKA pathway, such as 4E-BP1, S6K and CREB. Only the IHC pictures cannot confirm authors conclusions.

Author Response

The authors study the Haloperidol-induced immediate early genes in striatopallidal neurons with specific A2A and NMDA receptors relevant and via mTOR and PKA pathways. These investigations should be interesting.

We thank the reviewer for his/her encouraging message.

However, some major defects exist:

1, the authors only provide some IHC images as well as quantitative analysis based on IHC data. These observations are largely not enough to make solid conclusions. Western blot study and quantitative real time RT-PCR of IEG gene expression changes should be more convincing. Therefore the authors are request to perform western blot analysis and RT-PCR to ensure all IEG gene expression after Haloperidol treatment.

We understand this point but western blotting and RT-qPCR will not give us the cell type-specific information of induced IEGs. Previous studies have already shown this expression by RT-qPCR (Robbins et al., 2008 PMID: 18208916; Li et al., 2014 PMID: 15312167; Sakuma et al., 2015 PMID 25693194). The novelty of our study relies on the spatiocellular response of D2R-containing neurons which can be assessed only by IHC. 

2, The authors only observe some phenotype changes of Haloperidol treatment, such as A2A and NMDA receptors relevant and PKA, mTOR pathway relevant. However, what the logical links among these observations. So the study is very superficial with little impactful and novel knowledge. The potential impact of current study is very limited.

Within striatal MSNs, PKA and mTOR pathways are major players in the coincidence detection of neurochemical stimuli. It is well known that adenosine and glutamate, via the A2aR and NMDAR respectively, can scale the activity of these signaling cascades. Here, we describe that each pathway [at the extracellular (receptors) or intracellular (cascades)] is important but that the synergism between them is necessary for the responsiveness of striatopallidal neurons to haloperidol.

We believe that this is an important information given the fact that haloperidol and D2R antagonists are currently used as antipsychotic drugs.  

3, there are many defects on experimental design:

(1), in Figure 1, the authors should stain the tissue with hoechst so as to identify how many cells exposed in the image (D2R+ and D2R-). How the authors count the total D2R- cells? There are quite lots of black color holes in the images, which should be the cells without staining by red and green color. If only count the positive stained cells, that will be misleading. This is because the total numbers of D2R+ and D2R- cells are not the same. The author should study the percentage of red color cells of D2R+ and D2R- cells.

We believe that there is a misunderstanding.

To discriminate between D2R-positive and D2R-negative we used Drd2-eGFP mice, where eGFP is expressed under the promoter of the Drd2 gene. Therefore, all D2R-expressing neurons will appear in green. This helped us to visualize where IEGs were induced.

In Figure 1, we do not mention the percentage of colocalizing neurons since we believe that this information is not essential for the biological understanding of the manuscript as Arc and cFos are only triggered in D2R-positive (eGFP-positive) neurons (absolute not relative number of IEGs-expressing neurons) (see new Fig. 1).

(2), in Figure 4, the authors should study the phosphorylation status of downstream proteins of mTOR and PKA pathway, such as 4E-BP1, S6K and CREB. Only the IHC pictures cannot confirm authors conclusions.

We understand this point. However, it should be mentioned that the molecular components of these two signaling pathways following haloperidol administration have been already investigated in previous works mentioned in the manuscript. Here, we focused on IEGs which represent the long-term adaptation of events occurring in MSNs. The use of T34A and rapamycin supports the hypothesis that PKA and mTOR pathways are important but together necessary for the long-term responsiveness of striatopallidal neurons.    

Round 2

Reviewer 1 Report

the authors made all the necessary corrections to the text of the manuscript

Author Response

We thank Reviewer 1 and we are happy for his/her satisfaction regarding our work. 

Reviewer 2 Report

The revised version has not added any new data requested to correct their defects. So should be rejected

Author Response

We apologize if we did not succeed in satisfying the requests of Reviewer 2. 

In the new version of the manuscript we have detailed the methodology (lines 121-131) and rephrased some conclusions. We would like also to mention that the use of Drd2-eGFP mice is quite extensive within the basal ganglia community for discriminating between MSNs. Indeed, MSNs represent around 95% of striatal neurons with the remaining 5% representing non-MSNs. But we also show that haloperidol restricts cFos and Arc exclusively in Drd2-eGFP-positive neurons. Therefore, we genuinely believe that the use of other cellular markers (hoechst, DAPI for counterstaining) will not really change our take-home message.     

In the discussion we have also included a sentence (lines 332-334) calling for the use of modern modern high-throughput quantitative technologies to further studies the molecular adaptations and regulations induced by antipsychotics. 

As mentioned in our rebuttal letter we fear that the experiments suggested by Reviewer 2 will not add novelty to our work. Indeed, we agree on the use of more quantitative studies (critical point now included in our discussion), but western blotting and/or RT-qPCR will not be able to discriminate between cell types (whole striatal lysate/extracts). In addition, RT-qPCR will inform us on changes occurring at the transcriptional level, not necessarily on the functional proteins (cFos, Arc and Zif268). Again, these experiments have been already done in early studies. Moreover, including experiments with WB and RT-qPCR only for duplicative purposes will require the use of a large amount on animals which is against our European Directive 2010/63/EU (refinement, replacement and reduction). 

The same comment apply for the activation of cAMP/PKA and mTOR pathways. We have provided the essential literature which already shows that haloperidol triggers the activation of both cascades (our and other groups). 

We truly hope that Reviewer 2 will understand our argumentations.